# Pyrethrins and Pyrethroids: A Comprehensive Review of Natural Occurring Compounds and Their Synthetic Derivatives

**DOI:** 10.3390/plants12234022

**Published:** 2023-11-29

**Authors:** Camelia Hodoșan, Cerasela Elena Gîrd, Mihaela Violeta Ghica, Cristina-Elena Dinu-Pîrvu, Lucica Nistor, Iulius Sorin Bărbuică, Ștefan-Claudiu Marin, Alexandru Mihalache, Lăcrămioara Popa

**Affiliations:** 1Formative Science in Animal Breeding and Food Industry Department 59, Faculty of Engineering and Animal Production, University of Agronomic Sciences and Veterinary Medicine, Mărăști Bvd. District 1, 011464 Bucharest, Romania; cameliahodosan@yahoo.com (C.H.); lucia_mamina@yahoo.com (L.N.); sorin.barbuica@yahoo.com (I.S.B.); 2Department of Pharmacognosy, Phytochemistry and Phytotherapy, Faculty of Pharmacy, “Carol Davila” University of Medicine and Pharmacy, 6 Traian Vuia Str., 020950 Bucharest, Romania; 3Department of Physical and Colloidal Chemistry, Faculty of Pharmacy, “Carol Davila” University of Medicine and Pharmacy, 6 Traian Vuia Str., 020950 Bucharest, Romania; mihaela.ghica@umfcd.ro (M.V.G.); cristina.dinu@umfcd.ro (C.-E.D.-P.); stefan-claudiu.marin0720@stud.umfcd.ro (Ș.-C.M.); alexandru.mihalache0720@stud.umfcd.ro (A.M.); lacramioara.popa@umfcd.ro (L.P.)

**Keywords:** pyrethrins, pyrethroids, chemical structure, biosynthesis, biological properties, toxicity

## Abstract

This comprehensive scientific review provides an in-depth analysis of both the natural compounds, pyrethrins, and their synthetic derivatives, pyrethroids, focusing on their classification, biosynthesis, mechanism of action, general and pharmaceutical uses, as well as their toxicity and environmental impact. Pyrethrins, derived from certain plant species, have long been recognized for their potent insecticidal properties. The review begins by examining the classification of pyrethrins and pyrethroids, elucidating their structural characteristics and unique features within the field of natural and synthetic compounds. The biosynthetic pathways responsible for producing pyrethrins in plants are discussed, highlighting the enzymatic reactions and genetic regulation involved. In addition, the synthesis of pyrethroid derivatives is explored, including both natural and synthetic sources and potential optimization strategies. Understanding the mechanisms of action by which pyrethrins and pyrethroids exert their insecticidal effects is a crucial aspect of this review. Complex interactions with the nervous systems of target organisms are examined, providing insights into their selective toxicity and modes of action. In addition, the various applications of these compounds are explored, from their use in agriculture for pest control to their incorporation into household insecticides and potential pharmaceutical applications. The review also critically evaluates the potential toxicity of pyrethrins and pyrethroids to human health. By consolidating current knowledge and research findings, this review provides a comprehensive understanding of the properties and applications of pyrethrins and pyrethroids, highlighting their benefits and risks, and the importance of responsible and sustainable use in various areas.

## 1. Introduction

Botanical derivatives have served as fundamental pesticides for centuries and are renowned for their ecological properties [1,2]. Throughout history, various compounds such as sulfur, arsenic, and nicotine sulfate have been used to control pest infestations. After the Second World War, organochlorines and organophosphates emerged, marking a milestone in the development of pesticides. However, due to growing concerns about their environmental impact, alternative approaches such as integrated pest management (IPM) gained ground [3].

The complete ban on persistent organic pollutants, including DDT (dichlorodiphenyltrichloroethane), highlighted the need to re-evaluate pesticide use [4]. Subsequently, the advent of pyrethroids in the 1970s offered a promising alternative, offering efficacy and reduced environmental damage compared to their predecessors [5,6]. Despite their indisputable benefits in limiting agricultural losses and ensuring food security, the widespread use of pyrethroids and other pesticides has raised concerns about their negative effects on human health and the environment [7,8].

In recent years, new aspects of pyrethrins and pyrethroids have been revealed through research projects. For example, the field evaluation of novel spatial repellent controlled release devices against mosquitoes was studied in an outdoor setting in the northern Peruvian Amazonia [9]. Besides, a potential adjunctive agent for treating scabies was discovered: in vitro killing activity of permethrin and tea tree oil on *Sarcoptes scabiei* collected from patients. Parasites that were not damaged during sampling, and showed full motion were included [10]. Moreover, it was demonstrated for the first time the effect of pyrethroid exposure on the internal microbiota in *Aedes albopictus* [11].

In the environmental impact area, adverse effects of permethrin and tetramethrin in vitro models of freshwater mussels exposed to 1 mg/L, 10 μg/L, 100 ng/L, and 1 ng/L concentrations of chemicals for 24 h were also determined [12]. For this purpose, reduced glutathione activities were evaluated as biomarkers of the primary gill and digestive gland cell cultures [12]. To elucidate the interactive actions of pesticides on crop pollinators it was determined the individual and joint toxicities of thiamethoxam and other seven pesticides (dimethoate, methomyl, zeta-cypermethrin, cyfluthrin, permethrin, esfenvalerate, and tetraconazole) to honeybees (*Apis mellifera*) with feeding toxicity test [13]. In another study, the phenotypic and genotypic insecticide-resistance profiles among wild Anopheles were collected over 3 years to assess the longitudinal effects of dual-active-ingredient LLINs (long-lasting insecticidal nets) on insecticide resistance [14].

The application of pyrethroids and carbamates represents an environmental risk and may exert adverse effects on beneficial microorganisms such as *Trichoderma*, which contribute to the biocontrol of several fungal phytopathogens [15]. It was determined the influence of this insecticide on the release of enzymes such as chitinases, peroxidases, and endoglucanases by a consortium of selected *Trichoderma* strains grown in liquid culture medium) [15].

This review provides a comprehensive analysis of the origins, development, and implications of pyrethrins and pyrethroids. By examining the historical context and contemporary challenges associated with their use, we aim to provide a comprehensive understanding of the role and impact of these compounds in modern agriculture and public health. In addition, the review sheds light on the intricate mechanisms and comparative toxicological profiles of pyrethrins and pyrethroids in both mammals and insects, highlighting their variable sensitivity and implications for human and environmental health.

## 2. Natural Occurring Compounds—Pyrethrins

Plant materials containing pyrethrins have been traded in Europe since the middle of the 18th century. The species *Chrysanthemum cinerariaefolium* first appeared in the Dalmatia region of the Balkans and replaced a species originating from Persia, which proves that the true origin for the use of this plant is the Middle East, and dates to the 17th century. The first studies that set the foundation for the discovery of pyrethrins were elaborated in the 1920s, with the discovery of pyrethrins I and II [16]. These breakthroughs acted as a catalyst for the creation of a new market to sell the herb. The First World War also greatly influenced the countries that had a monopoly on the market, with Kenya becoming the leading exporter of plant products due to its higher concentration of active ingredients. Due to the pressures of World War II and other factors such as insect-borne diseases, these valuable pyrethrins were formulated as aerosols for their use against mosquitoes to decrease the transmission rate of malaria or yellow fever [16].

Another use of this class of plant compounds has been in agriculture as insecticides that are not highly toxic to mammals and are biodegradable. This led to their use on an industrial scale and the need to manufacture large quantities. In the production of pyrethrin-based insecticides, the starting point is generally the plant material, namely the mature inflorescences, which are dried and ground to a fine powder [2]. The powder can be used as such but can also be subjected to extraction with organic solvents.

Subsequently, different synthetic methods were developed for the 6 natural compounds, and new compounds structurally like the natural pyrethrins were created, thus generating a new class of compounds known as pyrethroids. These pyrethroids are superior to the natural compound due to their photostability, pyrethrins have a reduced photostability and are more biodegradable [17].

The term pyrethrum refers to the powdered and dried plant product extracted with organic solvents. In general, the term is used for the name of insecticide, either plant product or extract, which contains all the pyrethrins of this class of compounds. Several species are documented of which, after processing, the final product is called pyrethrum. The most famous species is *C. cinerariaefolium* (Dalmatian pyrethrum) due to its highest concentration of pyrethrins among all known species that produce this class of compounds, followed by *C. coccineum* (Persian pyrethrum) whose concentration in active principles is not as renowned [18], but their presence has been documented. Another species of the genus containing this class of compounds is C. balsamita also known as costmary [19]. Two other species from different genera that have been documented to contain pyrethrins are *Anacyclus pyrethrum* [20] and *Tagetes erecta* [21], all of which are members of the *Asteraceae* family.

### 2.1. Chrysanthemum cinerariaefolium

*Chrysanthemum cinerariaefolium* alias *Tanacetum cinerariaefolium* (Dalmatian pyrethrum) is the most industrially cultivated plant for pyrethrum production. The active ingredients are present in all organs of the plant but are concentrated in the inflorescences where, on the surface of the ovaries, there are secretory glandular hairs responsible for the abundant presence of pyrethrins in the plant tissues. Ref. [22] observed that the concentration of pyrethrins is higher in the epidermis of young leaves, and as the leaves mature the highest concentration of pyrethrins is found in the mesophyll of the leaf. This perennial species, since it is the most widely cultivated, has become the subject of most research papers based on the study of pyrethrins. From a morpho-anatomical point of view, the plant presents a capitulum-type inflorescence with central (tubular) yellow flowers and white radiating (ligulate) flowers, slender erect stems, and its leaves are alternate and pinnately lobed.

The existence of five new chemotypes within the species *C. cinerariaefolium* is documented from a sample of 25 populations in Croatia and distinguishes that the chemotype labeled “A” has the highest concentration of pyrethrin I. The percentage varies between these chemotypes from 0.36% of dry flower weight to 1.30% of dry flower weight. This finding opens the possibility of using this chemotype in the development of a new variety with higher concentrations of active compounds [23].

*C. cinerariaefolium* plantations usually are established in regions where the well-drained soil is rich in phosphorus, magnesium, and calcium, the average pH of the soil needs to be around 5.6, and, from a climatic point of view, a higher altitude where the precipitation needs to be minimum of 750 mm of rainfall distributed equally across the four seasons [24]. Large plantations and exports of *C. cinerariaefolium* are not present in Europe whereas one of the largest exporters of this insecticide is in Africa, namely Kenya [25], and also in Australia on the Tasmanian Island [26]. A newer agricultural industry is being developed in the highlands of Papua New Guinea, where the redeveloping of this plant culture offers a helping hand to improve the livelihoods of the local people [27].

### 2.2. Chemical Structures and Classification

Natural pyrethrins are the six ester compounds that result from the condensation of one alcohol and acid, the general structure is shown below (Figure 1). The acid half may be pyrethric acid or chrysanthemic acid, and the alcohol half may be one of the unsaturated hydroxy-cycloketones (rethrolones): cinerolone, jasmolone, and pyrethrolone. If the compound has pyrethric acid in its structure, then the three acid-alcohol combinations belong to category I of pyrethrins, and if chrysanthemic acid is replaced by pyrethric acid then the three remaining compounds belong to category II of pyrethrins (Figure 2).

### 2.3. Biosynthesis

Pyrethrins are the final product of two major pathways merging, as shown below (Figure 3), one pathway is responsible for the synthesis of the acid moiety (crysanthemic acid—CA and pyrethric acid—PA) and another pathway is responsible for the synthesis of the alcohol moiety or rethrolones (jasmolone, cinerolone and pyrethrolone). It has been revealed that most of the reactions involved in the biosynthesis of pyrethrins, in seedlings of *C. cinerariaefolium* take place in plastids [28]; however, it is also stated that part of these reactions (the alcohol moieties biosynthesis) also take place in the ER (endoplasmic reticulum) and in the peroxisome [29].

In the synthesis of the acid moiety, the starting compound is dimethylallyl diphosphate (DMAPP), which is the universal terpene precursor. Two molecules of the DMAPP are condensed resulting in chrysanthemyl diphosphate (CPP) and after eliminating the phosphate group by hydrolysis the final product of this step is chrysanthemol (CH-OL). This two-step reaction is catalyzed by the enzyme chrysanthemyl diphosphate synthase (CDS) also known as chrysanthemol synthase (CHS) [30], CPP is, because it is an irregular terpene due to the head-to-middle dimerization of DMAPP. CH-OL is converted into CA by two-step oxidation, firstly the alcohol is transformed into an aldehyde (chrysanthemal), catalyzed by *C. cinerariaefolium* alcohol dehydrogenase 2 (TcADH2), followed by an oxidation to CA, a reaction catalyzed by and aldehyde dehydrogenase 1 (TcALDH1) [31].

An alternative pathway that CH-OL can take is the production of the second acid moiety: PA. The previous two enzymes, TcADH2 and TcALDH1, which are responsible for the oxidation of the C1 hydroxyl group in CH-OL, “team up” with two other enzymes: chrysanthemol 10-hydroxylase (TcCHH), which is a cytochrome P450 oxidoreductase responsible for adding a carboxylic group to C10 after three oxidative processes; and 10-carboxychrysanthemic acid 10-methyltransferase (TcCCMT), a SABATH-class methyltransferase that adds a methyl group to the newly formed C10 carboxylic group [31].

The starting molecule from which all three alcohol moieties are synthesized is linoleic acid, which is converted to jasmonic acid (JA) by a series of consecutive reactions catalyzed by the following enzymes: *T*. *cinerariaefolium* lipoxygenase (TcLOX), allene oxidase synthase (TcAOS), allene oxide cyclase (TcAOC), cis (1)-12-oxo-phytodienenic acid reductase (TcOPR), succeeded by three _β_-oxidation reactions. The next step of converting JA into jasmone (JAM), which needs to pass more chemical reactions to become the alcohol moiety, has yet to be elucidated, although the recent conclusion of [32], states the fact that JA is not the only intermediate that produces the alcohol moiety, instead the precursor might also be 12-oxo-phytodienoic acid. The JAM is further hydroxylated by jasmone hydroxylase (TcJMH) into one of the final rethrolones, jasmolone (JAML) [33]. It has been shown that from JAML, the other two rethrolones can be converted, for example, the reaction of converting JAML into pyrethrolone (PYL) is catalyzed by *C. cinerariaefolium* cytochrome P450 family member, CYP82Q3 named Pyrethrolone synthase (TcPYS) [29]. Little to nothing is known about the third and last alcohol moiety, cinerolone (CINL), studies are yet to be conducted.

Finally, the end of biosynthesis of pyrethrins is the condensation of the two moieties, a reaction catalyzed by GDSL lipase-like protein enzyme, that takes place in apoplast [34], resulting in a new ester bond between the alcohol and the acid.

The major site of pyrethrin biosynthesis is the glandular trichomes located mainly on the surface of disc floret ovaries of *C. cinerariaefolium*, although these glandular trichomes, rarely may be present on other aerial parts of the plant. It is considered that these pyrethrin-producing species do not metabolize these kinds of compounds for their insecticidal proprieties nevertheless for self-protecting from herbivores [35]. Different research studies have exhibited that in field conditions, mechanical trauma put on plant individuals did not influence the total concentration of pyrethrins [36], whereas research studies done under specialized parameters (plants grown in growth chambers) have documented increased levels of pyrethrins in vegetative tissue of C. cinerariaefolium, hinting the fact that pyrethrin production may be also mediated by volatile organic compound (VOC) emitted by the plant from the wound location [37]. It is indicated that there is a difference between the levels of pyrethrin I and II in plant tissue from individuals who have been mechanically stressed compared to other individuals who have also been mechanically stressed but have been protected from VOCs secreted by the wound. This finding implies that the biosynthetic pathway of some pyrethrins may be triggered by external factors, such as VOCs, whereas some may be regulated by an internal systemic response [38].

### 2.4. Mechanism of Action

For both pyrethrins and pyrethroids the mechanism of action follows the same pattern, yet because of the biochemical engineering that led to the creation of pyrethroids, the synthetic class has a longer effect and a bigger potency. It has been determined, by molecular genetic studies done on houseflies centered on the resistance of pyrethroids [39], that the central site of insecticidal action revolves around the sodium voltage-dependent channels (SVDC). SVDC consists of a long transmembrane amino acid chain (Figure 4), with the amino and carboxy termini situated intracellular, forming two major subunits: α subunit (which is the main center of action) and a_β_ auxiliary subunit (responsible for the modulation of sodium channel gating and/or protein expression) [40]. The α subunit is composed of four main domains (I–IV), each domain has six transmembrane segments (S1–S2). The S4 contains positively charged amino acids and it acts as the voltage sensor when the membrane is depolarized receiving a confirmation charge that opens the channel. The channel is represented by the S5, S6, and the linkage segment between them, this segment also confers specificity to Na^+^ ions. There is an amino acid linkage chain between the domains III and IV which represents the inactivation gate. The channel can exist in four states due to the inactivation gate and one more site named the “activation gate”: the first state when the membrane potential is resting, the channel is closed but the inactivation gate is opened (closed state) when the depolarization is detected, the channel fully opens and the Na^+^ ions can freely pass thru (activated state). The inactivation state is initiated by the inactivation gate closing, clogging up the channel, after this step the channel finial closes and enters its deactivated state.

SVDC and potassium voltage-dependent channels (PVDC) are responsible for the movement of Na^+^ and K^+^ across the membrane which create action potentials, electrical impulses that travel along the neurons [41]. When a neurotransmitter binds to its receptor, in the synapse, a depolarization of the neuron membrane takes place and S4 segments of the SVDC detect the difference in charge, activating the channel and letting Na^+^ enter the neuron which leads to a further depolarizing of the membrane. This increasing concentration of Na^+^ oversees the rising phase of the action potential, leading to the membrane polarity switch. The following phase, the falling of the action potential, is produced by the inactivation and closing of the SVDC and the activation and opening of the PVDC, letting the K^+^ leave the neuron which leads to the hyperpolarization of the membrane. Finally, the PVDC closes and the membrane potential is back to its resting phase. All four stages of the ion channel can be viewed below (Figure 5).

The pyrethrins and pyrethroids lessen the peak of Na^+^ ions (the rising phase) by slowing the SVDC activation state [42,43], but also prolong the SVDC opening time by slowing the activated and inactivated state of the channel (Figure 6). Immediately after the fast absorption of the insecticide molecule through the insect’s cuticle, the neurotransmission fails causing a paralyzing state and possibly death in seconds or minutes.

The molecular structure and specific geometry enable pyrethrins and pyrethroids to bond to specific sites of the SVDC, mainly the S4–S5 linking segment of the II domain but other binding sites may be included too such as S6 of the III domain and S5 of the II domain, depending on the molecular structure of the insecticide. Regarding the specific geometry of the molecule, these insecticides must have a distinct stereospecificity, for example, all-natural pyrethrins have a 1R, trans configuration which grants the insecticidal action [44]. Moreover, the addition of an α-cyano substituent on the 3-phenoxybenzyl moiety (in different types of pyrethroids furthermore explained in the following chapters) greatly increases the insecticide action. Due to the diversity of the chemical structure of pyrethrins and pyrethroids, it is noted that a special type of chemical substituent or a specific reactive site cannot precisely justify insecticidal activity. For example, between Pyrethrin I and Etofenprox, there are little to nonchemical similarities; however, both molecules exude insecticidal activity. Yet, more details have been outlined about each segment of the pyrethrin molecule and comparisons of the different structures found in synthetic compounds [45,46].

Different mutations on the SVDC-expressing gene are responsible for substituting amino acids from distinct parts of the chain, this replacement leads to fewer binding sites for the insecticidal molecule causing insecticide resistance [47,48].

### 2.5. General and Pharmaceutical Uses

Since the newly developed pyrethroids have been on the market, the usage of pyrethrins in different forms for repelling insects has greatly decreased. Although the insecticidal power of pyrethrins and pyrethroids is recognized, they are generally associated with different insecticides such as seasmin, sulfoxide, and dicarboximide, for the synergetic potentiation of the effect [49]. Their primary use as a household insecticide is still noted today, different types of sprays, aerosols, solutions, and incense are used to prevent unwanted insects in the household.

A multitude of veterinary products containing this class of compounds are used for their properties to protect pets against insect pests, amongst these formulations can be found collars, powders, shampoos, baths, aerosols, sprays, etc. Among all these formulations for veterinary use, two categories of products can be distinguished, namely products using an organic solvent (alcohol or petroleum), which are more prone to toxic and/or adverse reactions, and products using a water-based solvent, whose bioavailability is relatively low.

The use of these compounds for human purposes has also been documented, namely permethrin for the treatment of various diseases caused by human ectoparasites, such as body lice, head lice, crab lice, and scabies [50]. Pyrethrins and pyrethroids have also been a great weapon against malaria, as mentioned before since the Second World War, these compounds have been used to repel mosquitoes carrying malaria. Nowadays impregnated be nets with different types of insecticides [51,52] are used in parts of the world where malaria Is common. These types of bed nets have been the focus of many debatesIse the long exposIre to insecticides may create different health issues due to the bioaccumulation in the human body [53,54]. Furthermore, it has been thoroughly investigated the resistance of mosquitoes to pyrethrin and pyrethroid-type insecticide [44,51,55], because of the rapid development of this type of phenomenon, especially on the African continent. In addition, it was pointed out the fact that a mixture of natural pyrethrins may be promising for malaria vector control, hinting at the fact that, in the long run maybe a constant juggling between different types of natural and synthetic compounds may be the key to tackle insecticide resistance to this class of compounds [56].

## 3. Pyrethrin Derivatives—Pyrethroids

### 3.1. General Considerations about Pyrethroids

Pyrethrin derivatives, known as pyrethroids, have a significant historical trajectory dating back to the mid-20th century, characterized by a series of crucial advancements in the realm of synthetic insecticide development. The milestone achievements include the synthesis of allethrin and bioallethrin in 1949, setting the stage for the subsequent introduction of the first-generation synthetic pyrethroid, resmethrin, in 1962, achieved through structural modifications of natural pyrethrins to enhance stability and elevate insecticidal efficacy [57,58].

The progression in this field expanded with the development of bioresmethrin in 1967, marking the commercial application of pyrethroids in the late 1960s, coinciding with the emergence of potent insecticides like cypermethrin and deltamethrin and solidifying the role of pyrethroids in the domain of pest management [59,60]. By 1983, the widespread utilization of pyrethroids encompassed over 33 million hectares annually, representing a significant share of the global insecticide market at 25.1% [3].

The recognition by the World Health Organization (WHO) of pyrethroids, particularly deltamethrin, and permethrin, for various applications, including the control of disease-transmitting mosquitoes through long-lasting insecticidal nets (LLINs), underscored their environmental advantages and reduced toxicity to humans and mammals [61]. This pivotal acknowledgment emphasized the increasing significance of pyrethroids in the context of global pest management strategies.

Categorized into type I and type II pyrethroids based on distinct chemical attributes influencing their insecticidal properties, these compounds have found extensive use across various sectors. Additionally, the incorporation of piperonyl butoxide as a synergist in commercial pyrethroid formulations has significantly contributed to their sustained efficacy in pest control [62,63].

Despite their pivotal role in pest control, concerns have surfaced regarding the impact of pyrethroids on non-target species and potential health implications for humans. Research efforts have focused on understanding specific toxicity biomarkers associated with pyrethroid exposure, shedding light on their adverse effects on various ecosystems [64,65,66,67,68].

Moreover, the escalating contamination of soil and water bodies with pyrethroids has spurred the exploration of eco-friendly remediation strategies to mitigate their environmental consequences [69,70]. Despite challenges and concerns, the versatility and effectiveness of pyrethroids have firmly established their position as integral components in the global insecticide market, finding applications in diverse sectors ranging from agriculture and forestry to household pest management [71,72]. This chapter aims to comprehensively examine the historical evolution, structural variations, applications, environmental implications, and health considerations associated with the use of pyrethrin derivatives, providing a holistic understanding of their significance and impact on contemporary pest management strategies.

### 3.2. Classification of Pyrethroids

Pyrethroid pesticides exhibit a clear classification into two distinctive categories, known as Type I and Type II, based on their behavioral toxicity and the presence or absence of an α-cyano group within their molecular structures [73]. This division is also reflected in the acute toxicity assessments, where most pyrethroids are grouped into classes I and II (Figure 7), indicating their respective levels of toxicity in rodent models.

Type I pyrethroids, which include allethrin, permethrin, resmethrin, bifenthrin, d-phenothrin, and tetramethrin, do not contain the α-cyano group in their chemical structure. As a result, they exhibit comparatively lower toxicity. In contrast, Type II pyrethroids, such as cypermethrin, deltamethrin, cyhalothrin (lambda), cyfluthrin, and fenvalerate (esfenvalerate), incorporate the α-cyano group, making them notably more toxic [58]. Type II pyrethroids have been associated with salivation, the choreoathetosis-salivation syndrome (CS), and motor dysfunction in mammals [71,72,73,74,75]. Additionally, various other effects, including oxidative stress, impacts on male fertility, and prenatal development, have been documented [66,75,76].

Human biomonitoring (HBM) studies typically monitor pyrethroid exposure through the detection of five specific metabolites in urine: 3-phenoxybenzoic acid (3-PBA), cis- and trans-3-(2,2-dichlorovinyl)-2,2-dimethylcyclopropane-1-carboxylic acid (cis-DCCA and trans-DCCA), 4-fluoro-3-phenoxybenzoic acid (F-PBA), and 3-(2,2-dibromovinyl)-2,2-dimethylcyclopropane carboxylic acid (DBCA). Among these, DBCA is specific to deltamethrin, and F-PBA is linked to cyfluthrin but can originate from both isomers. The metabolites are cis- and trans-DCCA are produced from the cis and trans isomers of permethrin, cypermethrin, and cyfluthrin, while 3-PBA is a common metabolite observed with various pyrethroids, including permethrin, cypermethrin, and deltamethrin [77].

However, the intrinsic susceptibility of these pyrethroids to sunlight-induced degradation prompted the development of next-generation compounds to address this issue. These newer compounds are categorized based on their photostability into two generations: first-generation and second-generation synthetic pyrethroids [72].

First-generation synthetic pyrethroids, including resmethrin, tetramethrin, and phenothrin, are derivatives of chrysanthemic acid esters, with susceptibility to photolysis, making them less stable [78]. In contrast, second-generation synthetic pyrethroids, such as permethrin, cypermethrin, and deltamethrin, exhibit both high potency and enhanced photostability, achieved through systemic modifications to their chemical structure in both the acid and alcohol moieties [78].

These pyrethroids share several common physical properties, such as low vapor pressure, low Henry’s law constants, higher octanol-water ratio coefficients (Kow), limited water solubility, enantiomers with identical physical properties, and diastereomers with differing physical attributes [78].

Synthetic pyrethroids have replaced natural pyrethrums due to their improved effectiveness and stability against insects, making them valuable tools for pest control. Various commercial synthetic pyrethroids are widely available, including bifenthrin, permethrin, deltamethrin, resmethrin, sumithrin, fenpropathrin, cyhalothrin, esfenvalerate, beta-cypermethrin, lambda-cyhalothrin, D-phenothrin, D-cyphenothrin, tetramethrin, and allethrin [79].

### 3.3. Synthesis Strategies of Pyrethroids: Approaches and Mechanistic Considerations

The synthesis of pesticides encompasses various strategies, each tailored to facilitate the creation of novel compounds with specific biological activities. Among the prominent methods, biomimetic synthesis simulates natural reactions within living organisms [80], while substructure splicing amalgamates active pesticide fragments to generate compounds with enhanced biological efficacy [81]. For the efficient synthesis of pyrethroids, the primary approach involves esterification, necessitating the initial production of chrysanthemic acid derivatives and alcohols (or aldehydes) [82,83,84,85].

Pyrethroids, renowned for their potent insecticidal properties and low mammalian toxicity, have evolved significantly, leveraging structural modifications to enhance stability and effectiveness [86]. The integration of aromatic groups into the alcoholic moiety has notably contributed to the improved stability of pyrethroids, allowing for their widespread application in crop protection [82,83,84,85]. This stability feature has propelled the exploration of various modifications, particularly in the alcohol and acid segments, underscoring the versatility of the synthesis approach [82,83,84,85].

### 3.4. Toxicological Insights into Pyrethroids: Human and Environmental Implications

Pyrethroids, a class of pesticides, have gained recognition for their relatively low toxicity in humans compared to other pesticide groups, including organochlorines, organophosphates, and carbamates [5]. However, type II pyrethroids have demonstrated greater acute oral toxicity than their counterparts, warranting a detailed assessment of their effects [5]. The toxicity profile of pyrethroids is diverse, with type I pyrethroids causing reversible skin and eye irritation upon exposure, while type II pyrethroids pose more severe risks due to their neurotoxic nature, often leading to fatal outcomes [5].

The human metabolism of pyrethroids involves various enzymes, including cytochrome P450s and carboxylesterases, contributing to the degradation of these compounds [87]. Despite their lower impact on humans relative to insects, pyrethroids can still induce alterations in various physiological functions, emphasizing the need for a comprehensive understanding of their toxicological effects [61].

#### 3.4.1. Toxicity in Humans

Pyrethroid poisoning primarily results from the disruption of sodium and chloride channels. Type I pyrethroids cause distinct symptoms known as type I syndrome, while type II pyrethroids, characterized by an additional cyan group in their chemical structure, elicit type II syndrome [88]. Instances of pyrethroid-induced cardiotoxicity have been reported, particularly associated with prallethrin, a common household pesticide used against mosquitoes, cockroaches, and houseflies [89].

Exposure to prallethrin has been linked to alterations in plasma biochemical profiles, with significant changes observed in glucose, phospholipids, nitrite, nitrate, and lipid peroxidase levels [90]. Additionally, allethrin and prallethrin exposure have been associated with increased MUC5AC expression in human airway cells and heightened reactive oxygen species production, underlining their potential impact on respiratory health [91].

Moreover, prallethrin poisoning has been implicated in gastrointestinal, respiratory, and nervous system disturbances, leading to metabolic acidosis and cardiac conduction disturbances [92]. Accidental and suicidal ingestions are the primary causes of pyrethroid poisoning in humans [93], with dermal exposure also being a common entry route [94,95,96].

In recent years, studies on pyrethroid toxicity in humans have increased considerably. As can be seen in Table 1, pathologies associated with pyrethrin exposure affect numerous systems and organs.

**Table 1 plants-12-04022-t001:** Health hazards of pyrethroids based on studies in recent years.

Subjects	Sampling	Exposure Assessment	Health Effects Related to Pyrethroid Exposure	Reference
512 pregnant woman	urine samples	urinary levels of cis-dibromodimethylvinylcyclopropane carboxylic acid (DBCA), 3-phenoxybenzoic acid (3-PBA), and total pyrethroid pesticides (PYR) metabolites	Exposure to PYRs during pregnancy was linked to higher birth weight, increased length at birth, and longer gestational duration, as well as a reduced likelihood of small for gestational age (SGA) or premature birth.	[97]
537 mother-child pairs	urine samples	dimethyl phosphate (DMP), dimethyl thiophosphate (DMTP), dimethyl dithiophosphate (DMDTP), diethyl phosphate (DEP), diethyl thiophosphate (DETP), diethyl dithiophosphate (DEDTP), 3,5,6-trichloro-2-pyridinol (TCPy; a metabolite of chlorpyrifos, chlorpyrifos-methyl and triclopyr), 3-PBA in urine	The findings of this research suggest that exposure to organophosphate and pyrethroid insecticides during pregnancy could potentially impact the normal growth of the fetus and lead to changes in birth anthropometric measures and gestational age, potentially signaling early disruptions in childhood developmental processes. Such effects might have long-term implications for individual health. Furthermore, our study emphasizes the differing susceptibility between genders to prenatal stressors. Given the limited evidence available at the time of this analysis and the discrepancies in the existing data, further investigation is strongly recommended to validate the reported outcomes and provide additional insights into the observed distinctions.	[98]
524 mother–child pairs	maternal urine samples	3-PBA in urine	Exposure to 3-PBA both during pregnancy and at age 2 was linked to heightened ADHD (attention deficit hyperactivity disorder) symptoms at age 6, whereas exposure during ages 4 and 6 was connected to ADHD symptoms at age 8. These results suggest the presence of various sensitive phases in early life that could influence the development of ADHD symptoms during the school-age period.	[99]
720 adolescents	urine samples	3-PBA in urine	Exposure to pyrethroid pesticides was found to be correlated with hearing loss in American teenagers. This research offers new insights into the connection between pyrethroid exposure and the sense of hearing.	[100]
1305 subjects	urine samples	2,2,3,3-tetramethylcyclopropanecarboxylic acid (TMCA), TFBA, 3-(2-chloro-3,3,3-trifluoroprop-1-enyl)-2,2-dimethylcyclopropanecarboxylic acid (CTFCA), 2,2-dimethyl-3-(2-methylprop-1-enyl)cyclopropanecarboxylic acid (MPCA), DCCA, FPBA, 3-PBA, MTFBL, and MMTFBL.	The findings of this study indicated that pyrethroid exposure was linked to either heightened levels of total prostate-specific antigen (PSA) or changes in the PSA ratio. Our findings propose that long-term exposure to pyrethroids could potentially harm male reproductive organs, particularly the prostate gland. Additionally, the effects of pyrethroid exposure on the body may vary depending on renal function status. The unstable statistical outcomes and the uncertainty surrounding the carcinogenicity of pyrethroids could be attributed to the limited sample size. Further comprehensive studies are required to establish the reproductive toxicity associated with pyrethroids.	[101]
2012 participants (1006 diabetic cases and 1006 matched controls)	serum samples	Serum pyrethroids determined by gas chromatography-mass spectrometry according to a standardized protocol—deltamethrin and fenvalerate	The comprehensive analysis of metabolites revealed that serum pyrethroid insecticides, especially deltamethrin, were linked to various plasma lipid metabolites, many of which participated in the glycerophospholipid metabolism pathway, predominantly characterized by PCs and LPCs. Additionally, the study indicated that four plasma metabolites (namely, PC 32:0, PC 34:4, CE 20:0, and TAG 52:5 [18:2]) and the pathway involving glycerophosphoethanolamine might represent the potential mechanism underlying the relationship between serum pyrethroids and the onset of type 2 diabetes (T2D).	[102]
177 children	urine samples	glyphosate (GLY); aminomethylphosphonic acid (AMPA); 3,5,6-trichloro-2-pyridinol (TCPy), the main chlorpyrifos pesticide metabolite; cis-(2,2-dibromovinyl)-2,2-dime thylcyclopropanecarboxylic acid (cis-DBCA); cis-3-(2,2-dichlorovinyl)- 2,2-dimethylcyclopropane-1-carboxylic acid (cis-DCCA); trans-3-(2,2- dichlorovinyl)-2,2-dimethylcyclopropane-1-carboxylic acid (transDCCA); 3-phenoxybenzoic acid (3-PBA); 4-fluoro-3-phenoxybenzoic acid (4-F-3-PBA); and cis-3-(2-chloro-3,3,3-trifluoroprop-1-enyl)-2,2- dimethylcyclopropanecarboxylic acid (CIF3CA or CFMP)	This study focused on monitoring children’s exposure to pesticides in Cyprus, following the methodology and tools used in the HBM4EU project. While a notable link was found between aminomethylphosphonic acid (AMPA) and the DNA oxidative stress marker in this children’s population, it is essential to replicate these findings in a more extensive study. The absence of significant associations between AMPA/GLY and lipid damage may suggest a biological DNA damage mechanism for AMPA.	[103]
683 mothers	maternal urine samples	cis-DBCA, cis-DCCA, trans-DCCA, and 3-PBA	Exposure during pregnancy to DDT and pyrethroid insecticides might be linked to concerning behaviors in children residing in a malaria-endemic rural area of South Africa.	[104]
726 adults	urine samples	3-PBA in urine	A direct correlation was noted between 3-PBA and shifts in both low-frequency and high-frequency hearing thresholds in individuals aged 20 to 39 in the United States, suggesting the susceptibility of young adults to the harmful effects of pyrethroid insecticides.	[105]
48 farmworkers	urine samples	five insecticide metabolites [3,5,6-trichloro-2- pyridinol (TCPy; a metabolite of the OP chlorpyrifos) and four metabolites of pyrethroid insecticides: 3-phenoxybenzoic acid (3PBA), 4-fluoro-3-phenoxybenzoic acid (4F3PBA), the sum of cis/trans 3-(2,2-dichlorovinyl)-2,2-dimethylcyclopropanecarboxylic acid (DCCA), and chloro-3,3,3-trifluoro-1-propene-1-yl (CFCA)]	The study showed a connection between exposure to OP and pyrethroid insecticides and a decrease in cortical brain activity in the prefrontal cortex, which might explain previously reported links to cognitive and behavioral function.	[106]
2116 adults	urine samples	3-PBA in urine	The results of this forward-looking group study revealed that environmental exposure to pyrethroid insecticides was notably linked to a heightened risk of mortality from all causes within the general adult population in the United States. The observed correlation is probably connected to the detrimental impact of pyrethroids on the cardiovascular system.	[107]
384 pregnant women	urine samples	PYRs metabolites: 3-phenoxybenzoic acid (3PBA), 4-fluoro-3-phenoxybenzoic acid (4F3PBA), and cis-2,2dibromovinyl-2,2-dimethylcyclopropane-1-carboxylic acid (DBCA) in urine samples	The results suggested that women in the final trimester of pregnancy and their infants, aged 6–8 months, were widely exposed to PYRs at low doses. The total concentration of PYRs metabolites in infancy exceeded that during pregnancy. While daily PYRs exposure during the third trimester of pregnancy did not impact toddlers’ language development, exposure to PYRs containing 4F3PBA and DBCA during the 6–8 months age range might delay toddlers’ language development. This demonstrates that the phase between 6–8 months old could be a sensitive window for PYRs exposure that affects toddlers’ language development. The likelihood of language development delay in 2-year-old toddlers could be anticipated by PYRs metabolites with 4F3PBA and DBCA during infancy, at 6–8 months.	[108]
1235 adults	urine samples	3-PBA in urine	The outcomes of our study revealed that exposure to pyrethroids was linked positively with testosterone (TT) and sex hormone-binding globulin (SHBG) in adults and negatively with circulating free testosterone in males. The current study provides backing to the idea that pyrethroids might have disrupted endocrine function on sex hormones at the observed exposure levels of pyrethroids in U.S. adults. Prolonged and chronic exposure to pesticides could potentially lead to changes in serum sex hormones.	[109]

#### 3.4.2. Biological Mechanisms and Environmental Impact

In both humans and animals, the nervous system serves as the primary target of pyrethroids, leading to acute neurobehavioral effects. The classification of pyrethroids into type I and type II groups is based on their specific neurotoxic manifestations in rodents and other species [109,110,111]. Environmental exposure to pyrethroids can be particularly harmful to aquatic life, emphasizing the need for stringent precautionary measures [112,113,114]. Similarly, pyrethrin and pyrethroid products can pose risks to avian species, particularly in the presence of certain carriers or propellants in spray formulations [115].

## 4. Conclusions

Pyrethrins and pyrethroids are a dominating group of insecticidal compounds that have been used for a long time and are still being used today, due to their potency and their variability. From natural pyrethrins, which can be utilized especially for their biodegradable properties, to the synthetic derivatives, pyrethroids, which may be used for their potency, this class of organic insecticides displays a lot of variability. It must be acknowledged that without plants, and plant metabolites, a great area of the insecticide compound class would be missing.

From the natural compound’s standpoint, the fully biosynthetic pathway of pyrethrins has yet to be elucidated, nonetheless clarifying the full path may be a key insight into genetically ingenerating subspecies of plants that may yield more pyrethrins, helping the pyrethrum industry flourish.

However, from the pyrethroid’s point of view, the constant demand for a new molecule that is less toxic, and more biodegradable, yet its potency does not lessen, may be an important catalyst to chemical engineering a compound that may satisfy all the necessities. The impetus of constantly developing and innovating the field of pyrethrins and pyrethroids also urges the research of the environmental impact of these compounds and the toxic effects on humans and animals, which in some cases may be fatal or threatening.

To conclude, this promising potential of pyrethrins and pyrethroids seems likely to persist in the future and needs constant innovation since all areas in which these insecticides are used, from agriculture, household insecticides, veterinary industry to the pharmaceutical and medical industry, require the development of new molecules or methods to analyze these compounds for different purposes.

## Figures and Tables

**Figure 1 plants-12-04022-f001:**
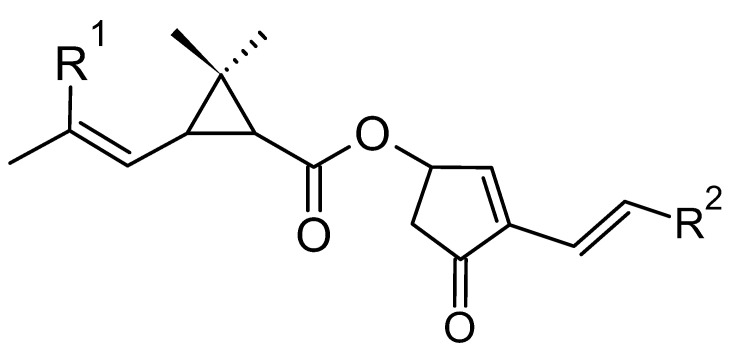
The general structure of pyrethrins.

**Figure 2 plants-12-04022-f002:**
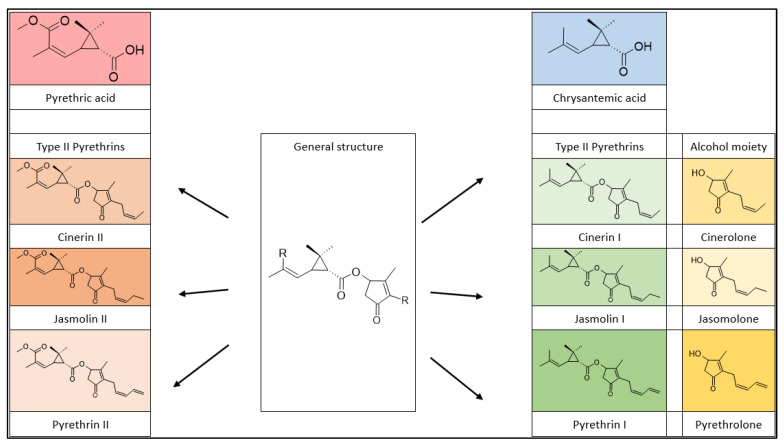
Pyrethrin classification.

**Figure 3 plants-12-04022-f003:**
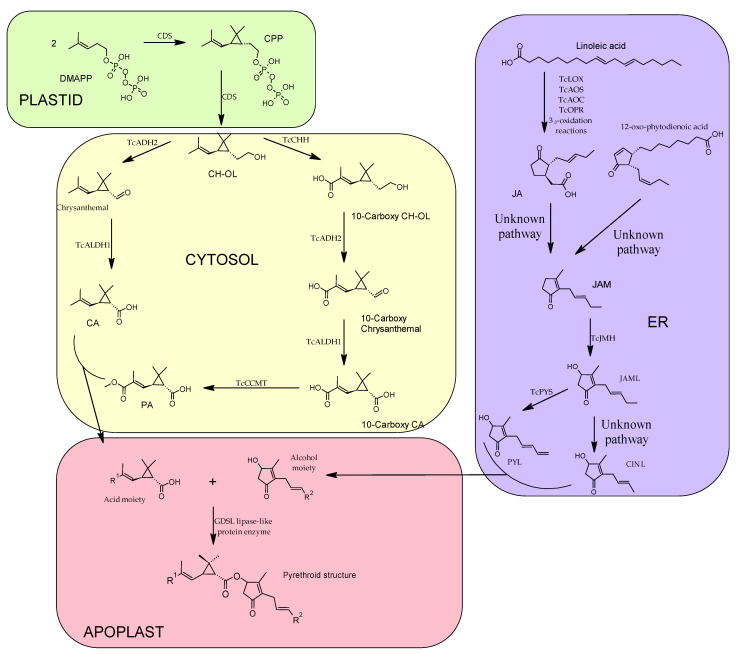
General biosynthetic pathway of pyrethrins depicting the cellular domains where the different reactions of the biosynthesis take place. DMAPP—dimethylallyl diphosphate; CDS—chrysanthemyl diphosphate synthase; CPP—chrysanthemyl diphosphate; CH-OL—chrysanthemol; TcADH2—*C. cinerariaefolium* alcohol dehydrogenase 2; TcALDH1—*C. cinerariaefolium* aldehyde dehydrogenase 1; CA—crysanthemic acid; PA—pyrethric acid; TcCHH—chrysanthemol 10-hydroxylase; TcCCMT—10-carboxychrysanthemic acid 10-methyltransferase; TcLOX—*T. cinerariaefolium* lipoxygenase; TcAOS—allene oxidase synthase; TcAOC—allene oxide cyclase; TcOPR—cis (1)-12-oxo-phytodienenic acid reductase; JA—jasmonic acid; JAM—jasmone; TcJMH—jasmone hydroxylase; JAML—jasmolone; TcPYS—Pyrethrolone synthase; PYL—pyrethrolone; CINL—cinerolone.

**Figure 4 plants-12-04022-f004:**
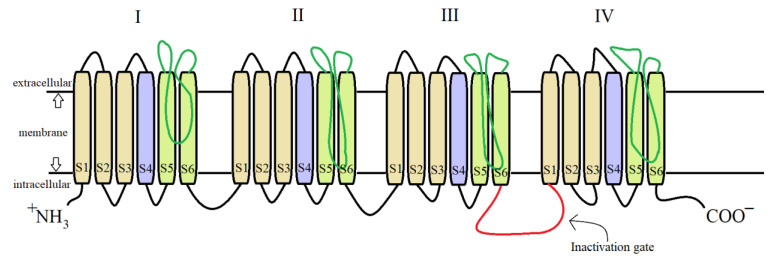
Detailed structure of the α subunit of the SVDC, the transmembrane amino acid chain with its four domains (I–IV), each of them having six transmembrane segments (S1–S6).

**Figure 5 plants-12-04022-f005:**
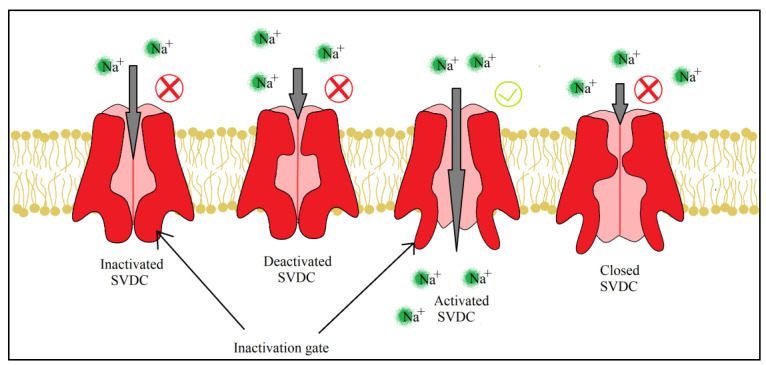
Pyrethroids and their effects on ion channels, the four states of SVDC, are dependent on the inactivation and activation gate. Readaptation of the representation from [41].

**Figure 6 plants-12-04022-f006:**
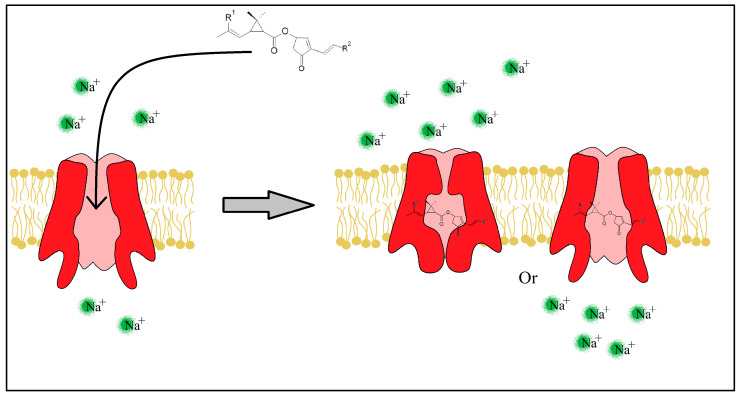
Insecticide molecule mechanism of action, by slowing the Inactivated or Activated state of the SVDC.

**Figure 7 plants-12-04022-f007:**
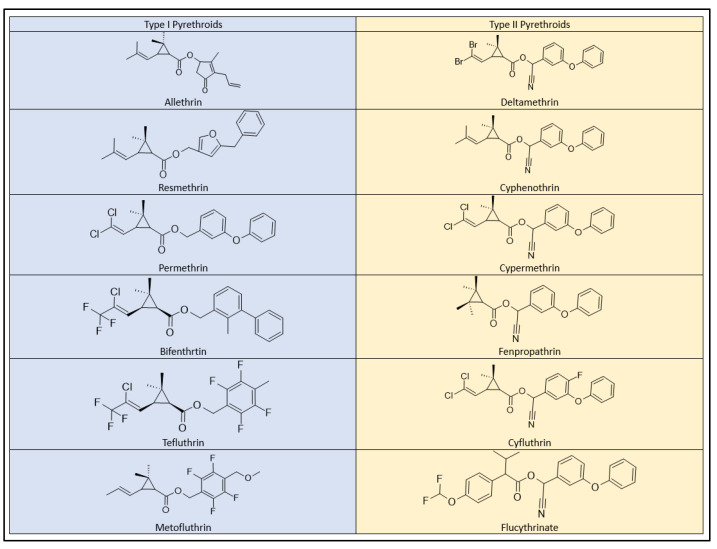
Type I and Type II pyrethroids structures.

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
