# Peer review of "Pyrethrins and Pyrethroids: A Comprehensive Review of Natural Occurring Compounds and Their Synthetic Derivatives"

_plants, 2023, doi:10.3390/plants12234022_

Round 1

Reviewer 1 Report

Comments and Suggestions for Authors

The manuscript entitled "Pyrethrins and Pyrethroids: A Comprehensive Review of Natural Occurring Compounds and Their Synthetic Derivatives" provides a review on the recent advances of pyrethrins and pyrethroids, considering their classification, biological sources, biosynthesis, mechanisms of action, and industrial applications. The manuscript is well-organized in adequate sections. Some questions and comments are stated below and should be addressed to improve the overall quality of the manuscript:

1. Keywords: Please consider revising for more meaningful and representative keywords of the review provided.

2. Introduction: The introduction should provide a more in-depth scientific background of the topics addressed in the review.

3. More references are needed to support the statements/evidences presented. For example, in this paragraph (lines 62-73) there is just one reference. In the next paragraph (lines 74-79), no reference was cited. Please revised it along the manuscript.

4. Please revise the format/style of references cited along the manuscript. For instance, in line 109, it is cited "Grdiša et al. 2013", but according to the journal guidelines, the authors should follow a numbered style (e.g., [1]) placed after the last name of the first author followed by "et al.". Please revise it along the manuscript.

5. The decimal numbers indicated in the manuscript must use point instead of comma to separate the decimal cases, for example, in line 112, replace "0,36%" by "0.36%", and "1,30%" by "1.30%".

6. Correct the word "classification" in line 115, please.

7.  Please cite the figures and tables in the manuscript before presenting the figures and tables.

8. Please define all abbreviations where they first appear in the manuscript. For instance, define the abbreviation "RE" in line 134.

9. Consider adding a list of all abbreviations used along the manuscript before the introduction.

10. In line 203, correct the number of the subtitle to "2.4." (instead of 2.3. that it seems repeated twice).

11. In lines 241-243, please reformat the reference presented in the Figure 5 caption following the journal guidelines.

12. In line 270, correct the number of the subtitle to "2.5." (instead of 2.4.).

13. In Table 1, please consider placing the reference column as the last column of the table. What is it the meaning of "1" after "Exposure assessment" in the fourth column of the Table 1? Please clarify it.

14. References: Please revise all references presented in the list following the journal guidelines.

Comments on the Quality of English Language

The manuscript may benefit from some revisions regarding minor English grammar/spelling mistakes.

Author Response

Dear Reviewer, 1

Thanks for making the significant effort of revising the manuscript. We appreciated your opinions and helpful suggestions. The current draft was significantly modified, being more focused on the subject. We hope you’ll find it better organized, according with your appreciations.

Please find hereafter our response to your indications. Modifications have been shown in red color in the main text, using Track changes.

Best regards,

The authors

  1. English language and style

( ) Extensive editing of English language and style required
( ) Moderate English changes required
(x) English language and style are fine/minor spell check required
( ) I don't feel qualified to judge about the English language and style

A thorough revision of English language and style was performed for the manuscript and all the corrections were made with track changes.

  1. Reviewer’s suggestion and authors’ answer

Reviewer 1: 1) Keywords: Please consider revising for more meaningful and representative keywords of the review provided.

Authors: In accordance with the reviewer’s indication we modified the keywords.

Reviewer 1: 2) Introduction: The introduction should provide a more in-depth scientific background of the topics addressed in the review.

Authors: In accordance with the reviewer’s indication, we added more in-depth scientific background of the topics (lines 55-79).

Reviewer 1: 3) More references are needed to support the statements/evidence presented. For example, in this paragraph (lines 62-73) there is just one reference. In the next paragraph (lines 74-79), no reference was cited. Please revise it along with the manuscript.

Authors: In accordance with the reviewer’s indication, we reviewed and adjusted the references in those specific paragraphs.

Reviewer 1: 4) Please revise the format/style of references cited along the manuscript. For instance, in line 109, it is cited „Grdiša et al. 2013” but according to the journal guidelines, the authors should follow a numbered style (e.g., [1]) placed after the last name of the first author followed by "et al.". Please revise it along the manuscript.

Authors: In accordance with the reviewer’s indication we changed the format of those specific references to suit the guidelines (lines 135, 217, 303, 335).

Reviewer 1: 5) The decimal numbers indicated in the manuscript must use point instead of comma to separate the decimal cases, for example, in line 112, replace „0,36%”by „0.36%”, and „1,3%” by „1.30%”.

Authors: In accordance with the reviewer’s indication, we swaped ’’,’’ with’’.’’ (line 138).

Reviewer 1: 6) Correct the word "classification" in line 115, please.

Authors: In accordance with the reviewer’s indication we correct the word.

Reviewer 1: 7) Please cite the figures and tables in the manuscript before presenting the figures and tables.

Authors: In accordance with the reviewer’s indication, we adjusted the citation of the figures and table (lines 152, 157, 168, 248, 277, 284).

Reviewer 1: 8) Please define all abbreviations where they first appear in the manuscript. For instance, define the abbreviation „RE” in line 134.

Authors: In accordance with the reviewer’s indication, we defined all the abbreviations (lines 48, 169,).

Reviewer 1: 9) In line 203, correct the number of the subtitle to "2.4." (instead of 2.3. that it seems repeated twice).

Authors: In accordance with the reviewer’s indication we corrected that error.

Reviewer 1: 10) In lines 241-243, please reformat the reference presented in the Figure 5 caption following the journal guidelines.

Authors: In accordance with the reviewer’s indication, we reformatted the reference.

Reviewer 1: 11) In line 270, correct the number of the subtitle to „2.5” (instead of 2.4.).

Authors: In accordance with the reviewer’s indication, we corrected the error.

Reviewer 1: 12) In Table 1, please consider placing the reference column as the last column of the table. What is it the meaning of „1”; after „Exposure assessment” in the fourth column of the Table 1? Please clarify it.

Authors: In accordance with the reviewer’s indication, we placed the reference column last and deleted „1” after the „Exposure assessment”.

Reviewer 1: 13) References: Please revise all references presented in the list following the journal guidelines.

Authors: In accordance with the reviewer’s indication, we revised all the references to fit the journal guidelines.

Reviewer 2 Report

Comments and Suggestions for Authors

Pyrethrins and pyrethroids – are pesticides found naturally in some chrysanthemum flowers. They are a mixture of six chemicals that are toxic to insects. These natural components are commonly used to control mosquitoes, fleas, flies, moths, ants, and many other pests. There are generally separated from the flowers of plant species: Chrysanthemum cinerariaefolium However, they typically contain impurities from the flower. Whole, crushed flowers are known as pyrethrum powder.

Pyrethrins and pyrethroids have been registered for use in pesticides since the 1950’s. They have since been used as models to produce longer lasting chemicals called pyrethroids, which are man-made.

Currently, these counpunds are found in over 2,000 registered pesticide products. Many of these are used in and around buildings and on crops and ornamental plants. Others are used on certain pets and livestock. They are commonly found in foggers, sprays, dusts and pet shampoos. Some of these products can be used in organic agriculture.

In regard to the environment, pyrethrins respectively  pyrethroids also stick to soil and have a very low potential to move through soil towards ground water. In field studies, pyrethrins were not found below a soil depth of 15 centimeters. However, pyrethrins can enter water through soil erosion or drift. In the top layers of soil, pyrethrins are rapidly broken down by microbes. Soil half-lives of 2.2 to 9.5 days have been reported. These chemical structures have a low potential to become vapor in the air.

From the above text and facts about these natural substances, I appreciate the presentation and manuscript of the authors from several research institutions in Bucharest, Romania. As part of the overview of this research fields with the classification of pyrethrins and pyrethroids, elucidating their structural characteristics and unique features within the 23 field of natural and synthetic compounds are presented. The very important are the explanation of the biosynthetic pathways and understanding the mechanisms of action by 27 which natural chemicals exert their insecticidal effects.

However, the text of the manuscript lacks specific information about growing of Chrysanthemum (large-scale cultivation), harvesting, post-harvest technologies (large-capacity extractions and standardization of insecticide preparations).

Comments on the Quality of English Language:  Minor editing of English language is needed.  It'd be of value if you had the manuscript proofread by a native English person.

Comments on the Quality of English Language

Minor editing of English language is needed.  It'd be of value if you had the manuscript proofread by a native English person.

Author Response

Dear Reviewer 2,

Thanks for making the earnest effort of revising the manuscript. We appreciated your opinions and helpful suggestions. The current draft was significantly modified and has much better flow and readability, being more focused on the subject. We hope you’ll find it better organized, according with your appreciations.

Please find hereafter our responses to your indications. A detailed response to each point raised has been shown in red colour in the main text, using Track changes.

Best regards,

The authors

  1. English language and style

(x) Extensive editing of English language and style required
( ) Moderate English changes required
( ) English language and style are fine/minor spell check required
( ) I don't feel qualified to judge about the English language and style

A thorough revision of English language and style was performed for the manuscript and all the corrections are made with track changes.

  1. Reviewer’s suggestions and authors’ answers

„However, the text of the manuscript lacks specific information about growing of Chrysanthemum (large-scale cultivation), harvesting, post-harvest technologies (large-capacity extractions and standardization of insecticide preparations). Comments on the Quality of English Language:  Minor editing of English language is needed.  It’d be of value if you had the manuscript proofread by a native English person. As a result, it is suggested that the novelty of the current study be emphasized more in the introduction section.

In accordance with the reviewer’s indication, we provided more information about the agricultural process of growing C. cinerariaefolium (lines 141-149) and had a thorough revision of English language for the publication.
